# Solid-State Cold Spray Additive Manufacturing of Ni-Based Superalloys: Processing–Microstructure–Property Relationships

**DOI:** 10.3390/ma16072765

**Published:** 2023-03-30

**Authors:** Alessandro M. Ralls, Mohammadreza Daroonparvar, Merbin John, Soumya Sikdar, Pradeep L. Menezes

**Affiliations:** 1Department of Mechanical Engineering, University of Nevada Reno, Reno, NV 89557, USA; alessandroralls@nevada.unr.edu (A.M.R.);; 2ASB Industries, Research and Development Department, Barberton, OH 44203, USA

**Keywords:** cold spray, Ni-based superalloys, additive manufacturing, tribology, oxidation, mechanical properties

## Abstract

Ni-based superalloys have been extensively employed in the aerospace field because of their excellent thermal and mechanical stabilities at high temperatures. With these advantages, many sought to study the influence of fusion-reliant additive manufacturing (AM) techniques for part fabrication/reparation. However, their fabrication presents many problems related to the melting and solidification defects from the feedstock material. Such defects consist of oxidation, inclusions, hot tearing, cracking, and elemental segregation. Consequentially, these defects created a need to discover an AM technique that can mitigate these disadvantages. The cold spray (CS) process is one additive technique that can mitigate these issues. This is largely due to its cost-effectiveness, low temperature, and fast and clean deposition process. However, its effectiveness for Ni-based superalloy fabrication and its structural performance has yet to be determined. This review aimed to fill this knowledge gap in two different ways. First, the advantages of CS technology for Ni-based superalloys compared with thermal-reliant AM techniques are briefly discussed. Second, the processing–structure–property relationships of these deposits are elucidated from microstructural, mechanical, and tribological (from low to high temperatures) perspectives. Considering the porous and brittle defects of CS coatings, a comprehensive review of the post-processing techniques for CS-fabricated Ni superalloys is also introduced. Based on this knowledge, the key structure-property mechanisms of CS Ni superalloys are elucidated with suggestions on how knowledge gaps in the field can be filled in the near future.

## 1. Introduction

In recent years, the development of additive manufacturing (AM) has attracted many due to its simplistic and rapid fabrication of complex components [1,2]. Compared with traditional manufacturing processes, AM has become increasingly used, as the standard costs of using external tools (coolants, cutting tools, etc.), physical labor, and material consumption (which is greater for traditional manufacturing methods due to the subtractive practices used) can largely reduce the profits of many firms [3,4,5]. This is especially true for industries that manufacture small-to-medium batches of products, as AM can help to reduce lead times and allow for the fabrication of new and intricate parts, as seen in the aviation, marine, and automation industries [6]. Examples of such commonly used AM techniques include fused deposition modeling (FDM), directed energy deposition (DED), friction stir additive manufacturing (FSAM), 3D electrodeposition, and selective laser sintering (SLS) [7,8,9,10,11,12,13]. In fact, many sources in the literature evaluated the cost differences between traditional manufacturing techniques to standard AM techniques and have all reached the same fundamental conclusion that AM is much more efficient from time, quality, and cost perspectives [14,15,16,17]. In addition to the operating cost advantage of AM techniques, AM also offers a wide array of advantages from a material performance perspective. This is largely due to the ability to control the final microstructure through varying the many different processing parameters pertaining to AM [10,18].

Based on these points, there are a variety of material systems that are widely used for AM, spanning from polymers to soft and hard metals [2,19]. Out of the existing variety of materials, the practice of fabricating Ni-based superalloys has attracted a great amount of attention in recent years [20]. This attraction can be largely attributed to their ability to maintain their strength (without suffering from creep and fatigue) and corrosion resistance (due to oxidation) at elevated temperatures (>540 °C) [20,21]. Particularly, this ability stems from the formation of protective Cr_2_O_3_ and Al_2_O_3_ scales along the surface, which can effectively assist with mechanical, tribological, and corrosion-related degradation (as shown in Figure 1) [22]. From an industrial perspective, these advantages are critical, as they can assist with the longevity of various components that are continually exposed to high stress and oxidative conditions [23,24]. One of the most common uses for AM-fabricated Ni-based superalloys is in the aerospace industry, in which standardized components, such as gas turbine disks, shafts, blades, and exhaust systems, are exposed to extremely high temperatures [25]. Among the various specific case studies, some of the more contemporary reports of applied AM-based Ni superalloys are from NASA, of which rocket propulsion systems for space rocket launches were successfully fabricated [26,27,28,29]. Additionally, others also reported the usage of AM-based Ni superalloys as thermal barrier coatings for other high-temperature applications [30]. That being said, although the application of AM Ni-based superalloys is largely utilized in aerospace, other industries, such as the automotive and nuclear industries, also see their usage [24].

The most frequent AM technologies that are used to fabricate Ni-based superalloys tend to be fusion- or thermal-based. For fusion-based AM, techniques such as powder bed fusion (PBF) and DED are most frequently used [24,32,33]. Fundamentally, these techniques share the common characteristic of melting material in an additive fashion. However, they differ in the aspect that PBF techniques rely on the melting of powders along a powder bed due to a high-intensity laser (or electron) beam [34,35,36] whereas DED actively adds the material (either in a form of a wire or powder) with the energy source [37]. In the case of thermal-based techniques, thermal spraying is most commonly used. In contrast to fusion-based AM technologies, the fundamental basis for thermal spray techniques mostly lies in the splat formation and solidification of rapidly accelerated molten powders. To briefly conceptualize the essence of thermal spraying, a feedstock material is heated to a molten-like state by an external heat source. Once heated, the feedstock material is then rapidly propelled to the surface of a substrate, of which the material bonds via the combination of thermal and kinetic energies (e.g., such as in the high-velocity oxygen fuel (HVOF) method). Upon solidification, a protective coating is formed upon the substrate material [38].

Although these techniques are thoroughly tested, they tend to suffer from defects pertaining to tensile stresses, oxidation, porosity, chemical composition inhomogeneity, and undesirable phase transformations from the molten pool/molten particles [10,39,40,41,42,43,44,45]. In the case of Ni-based superalloys, the formation of precipitates in the γ’ phase can result in hot cracking due to the inhibition of liquid feeding throughout the melting process [33,46]. The implications of such defects can result in premature part and component failures (especially in mechanical, wear, or corrosion-based environments), which can significantly result in exacerbated repair costs and decreased machine efficiencies. As such, there is an ever-increasing need to utilize a rapid non-thermal-based AM technique that can produce reliable and robust Ni-based superalloys.

One recent technology that has attracted the attention of nickel superalloy AM is cold spray additive manufacturing (CSAM) [47]. Acting as a solid-state technique, the fundamental mechanism of CSAM lies in the ability to form robust components without the need for thermal melting (as seen in fusion-based AM). This is achieved by rapidly accelerating micron-sized particles (typically between 5 and 50 μm) through an easily adjustable De-Laval (convergent–divergent) nozzle to a desired surface. In a sense, CSAM can also be utilized as an additive coating technique that can protect key components from degradation. The method for CSAM particle acceleration uses a pressurized inert gas, typically in the form of He or N_2_ [48,49]. Upon impact, any pre-existing oxides (either from the powder particle or sprayed surface) are effectively removed (Figure 2), which can result in physiochemical and mechanical bonding, thus resulting in a deposited layer [50,51,52]. Over time, these layers compressively form to create an AM build while preserving the original phases of the particle feedstock. The advantage of such a practice is that the typical defects of oxidation, grain growth, thermal stresses (in the form of tensile stresses), and phase transformations can be effectively avoided, in contrast to fusion-based techniques, such as DED and PBF [53]. From a deposition rate perspective, CSAM also yields the advantage of having greater deposition rates (up to 50 kg/h [54]) compared with fusion-based AM processes (which were reported to be up to 10 kg/h with processes such as DED) [55,56,57]), which can be visually seen in Figure 3. For reference, the fusion-based AM processes listed pertain to PBF-related (consisting of sintering and full melting) and DED-related (consisting of a laser beam, electron beam, and arc plasma) techniques [58]. Additionally, CSAM can also allow for the fabrication of components in the millimeter range [59,60]. However, it should be mentioned that brittleness from the immense compressive stresses can lead to premature crack propagation due to the immense cold working and high dislocation generation from the deposition. Additionally, porosity can also occur, which can serve as high-stress-concentration sites in mechanical/abrasive applications, which can further degrade the durability of Ni-based CSAM coatings [61,62,63,64]. However, post-processing techniques can effectively mitigate these defects. For example, one widely used group of post-treatments is heat treatment (HT), which effectively serves to densify/relieve the extreme compressive stresses of CS deposits. While used in various forms (e.g., annealing, hot isostatic pressing, and aging), the fundamental premise for HT is to allow for interparticle diffusion, followed by recrystallization and grain growth. Although there are many other treatments, such as laser melting, friction stir processing, and hot rolling, HT is one of the most utilized techniques used for CS deposits [63].

When considering the feasibility of the cold spray (CS) of Ni-based superalloys, the most critical factor for successful deposition depends on the “quality” of particle plastic deformation. This quality is essentially determined by whether the particle exceeds the critical velocity parameter associated with the material [65]. For Ni-based superalloys (such as IN 625), the critical particle velocity was determined to be 675 m/s, which can easily be achieved by adjusting the gas pressure [66]. Consequentially, due to the ease of deposition, there has been great attraction toward the utilization of CS for Ni-based superalloys over commonly used AM techniques. For example, shifts in the aerospace sector have allowed for the formation of the new AMS7057 standard to ensure proper quality control for CSAM-related components [67]. These components can largely be applied as blades, casings, thrust reversers, and aerodynamically heated skins [24]. Similarly, other industries, such as the automotive industry (for engine block protection), also began to widely utilize CSAM technology for Ni-based superalloys [68].

Although literature pertaining to the application of CSAM of Ni-based superalloys has been increasing, there is yet to exist a comprehensive understanding of their critical processing–microstructure–property relationships; especially considering general the push toward CSAM (alongside the large amount applications of which Ni-based superalloys can be used for), there is yet to exist a review that describes how the quality of CSAM Ni-based superalloys can be effectively tailored. Such an understanding can consequentially have immense industrial implications, as the lifespans of various moving mechanical assemblies (MMAs) can be extended, thus resulting in fewer occurrences of part replacement. The present review aimed to correlate the structural-, mechanical-, tribological-, and corrosion-based properties of Ni-based superalloys fabricated via CSAM, considering both the as-built and post-processed conditions/states. By doing so, this newly found understanding will allow for a larger scientific discussion of the possibilities of CS for Ni-based superalloys.

## 2. Structural and Microstructural Evolution

By understanding the microstructural forming mechanisms of CS with respect to Ni-based superalloys, the microstructure can be further controlled, thus manipulating the properties of the CS Ni-based superalloy deposits. Typically speaking, the degree of plastic deformation from the particle impact will largely dictate the structural and microstructural characteristics of CS coatings. However, most CS-related works tend to focus on materials such as copper and aluminum due to their ease of deposition. However, when compared with Ni-based superalloys, their structural characteristics are inherently different. As such, it can be expected that their deformation and recrystallization mechanisms will also differ [69].

Chaudhuri et al. [65] are among a few who thoroughly investigated the microstructural forming mechanisms of Ni-based superalloys and showed the viability of deposition. In their work, they primarily focused on the application of CS of IN 625 coatings on 4130 low alloy steel. For CS deposition, the authors used a commercial powder of IN 625 with a particle size of 5 µm to 50 µm. The high-pressure spraying system utilized helium as a propellant and pressurized/heated the gas to 30 bar and 500 °C. According to their findings, they were able to successfully deposit a 3 mm-thick coating on the substrate. From an interfacial standpoint, a significant amount of plastic strain developed near the interface on the substrate. This finding was reflected by the visually flattened particles, thus suggesting that sufficient plastic deformation took place. From a microstructural perspective, the degree of crystalline refinement was the greatest at the interface. These findings were also supported by the gradual decrease in hardness from the interface. The authors explained this phenomenon in terms of the accumulation of dislocations due to the severe plastic strain from the continual peening-like effect from the CS particles to the initially formed layers.

To further understand the underlying effects, the increase in dislocation density (due to the continual peening-like effect of the plastically deformed particles) eventually led to the generation of dislocation tangles and dislocation walls, as shown in Figure 4. As such, when the dislocation density of the IN 625 coating reaches a threshold value, it forms low-angle grain boundaries (LAGBs) due to the combined effect of dislocation annihilation and rearrangement. In a sense, the formation of LAGBs can be correlated with the combination of lessened deformation (strain) and adiabatic heat rise that occurs with geometrically uneven particles throughout the deposition process, thus leading to limited recrystallization [70,71]. However, due to the accumulated strain and increase in adiabatic temperature rise (from the following particles), the LAGB was effectively transformed into high-angle grain boundaries (HABGs).

Similar observations were also made throughout the inner microstructure of the coating, as a mixture of LAGBs and HAGBs was also observed within and around the interparticle splats (Figure 5). Due to the nature of the process, it can be inferred that the layers formed for IN 625 experience a degree of progressive lattice rotation due to the phenomena of continuous dynamic recrystallization (CDRX). In comparison to traditional fusion-based techniques (such as DED, which can also experience CDRX [72]), this mechanism was found to be advantageous in the sense that more relatively homogeneous microstructures can be achieved using CSAM [73,74]. As such, this phenomenon validates the feasibility of CS for Ni-based superalloys relative to traditional fusion-based techniques.
Figure 5(**a**) Inverse pole figure (IPF) map of the splat and splat boundaries in CS IN 625 coating microstructures. A noticeable grain fragmentation caused by high-strain-rate deformation was observed for both the splat and splat boundaries, (**b**) low-angle (in grey color) and high-angle (in black color) boundaries related to the splats (regions 1 and 3) and splat boundaries (regions 2 and 4). Reprinted from Chaudhuri et al. [65], Copyright 2017, with permission from Elsevier.
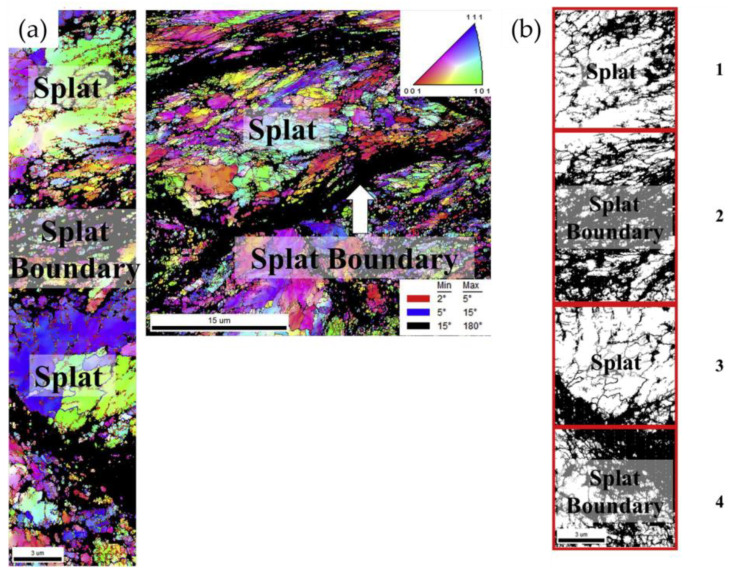



However, despite knowing that the CS technique is viable, the specific factors that influence this viability should be understood. Particularly, these factors are related to the particle impact velocity. As aforementioned, for sufficient deposition to take place, a particle must exceed a specific velocity. This factor is known as the critical velocity, which will essentially result in mechanical bonding to the substrate. However, past a certain point increasing the particle velocity will be detrimental to a successful deposition, which is also commonly referred to as the erosion velocity. Within this range is known as the window of deposition, which represents the ideal velocity at which a particle should be to obtain a successful deposition. To understand this relationship for Ni superalloys, Wu et al. [75] studied the varying deposition characteristics of high-pressure deposited IN 625 on an aluminum 6061 substrate in relation to their critical and erosion velocity. In their experiments, it was found that not only did the particle temperature affect the critical and erosion velocity but so did the particle size. As shown in Figure 6, the erosion velocity appears to decrease as the particle temperature increases. This can largely be attributed to the lesser degree of mechanical interlocking that occurs. Similarly, it can be seen that by increasing the particle size, the particle velocity decreases. As to be expected, this decrease can be attributed to the increase in physical aspects (i.e., mass) of the rapidly accelerated particles as per the conservation of kinetic energy ($E=\frac{mv^{2}}{2}$). However, interestingly enough, the particle temperature can also have some type of effect on the window of deposition. Specifically, by increasing the particle temperature, the window of the deposition begins to gradually decrease. This decrease is due to a type of thermal softening effect, in which there is a lesser reliance on particle mechanical interlocking [76].

Aside from these factors, other variables, such as the substrate surface finish and substrate pre-heating, can also alter this window of variation, as was found in the work of Sun et al. [77]. It was found that there was a general trend of increasing adhesion strength as the substrate surface roughness decreased due to the greater area of contact. Similarly, increasing the substrate pre-heating temperature allowed for greater adhesion due to the increase in local adiabatic heat upon impact. Due to this phenomenon, greater plastic dissipation occurs while elastic strain energy decreases.

Having the underlying knowledge that particle size/temperature and substrate finish/pre-heating can affect the window of deposition for IN 718, it is also important to consider spraying variables that can affect the depositing quality. The key factors consist of varying the type of inert gas alongside its pressure. Although temperature can have an influence, it is not nearly as influential as varying the gas pressure. Ma et al. [78] are among the few that studied the influence of various gases with respect to their pressure for IN 718. In their work, nitrogen and helium were utilized. They used three pressures, namely, 3 MPa, 5 MPa, and 7 MPa, and three temperatures, namely, 600 °C, 800 °C, and 1000 °C, for nitrogen. The pressure and temperature for the helium propellant gas were 3 MPa and 1000 °C, respectively. The authors observed significant severe plastic deformation (SPD) with the rise in nitrogen gas pressure with the helium propellant gas showing overall enhanced SPD and particle jetting compared with nitrogen. This can be largely attributed to the difference in weights for each gas. For better visualization, the splat morphologies of the deposited powder particles are represented in Figure 7a–d. As can be seen, the degree of jetting gradually increases with higher gas temperatures (for the N_2_ gas), which suggests that enhanced metallurgical bonding occurs due to oxide film cleaning.

From a cross-sectional perspective, the deposits fabricated using both processing gases resulted in dendritic structures at the interparticle boundaries, as indicated by the red arrows in Figure 8a,b. Although the authors did not explicate the origin of their formation, they were likely due to the microstructure of the initial powder used in the powder fabrication technique, especially since their formation was found mainly at the central regions of the particle in contrast to the severely deformed interparticle boundaries. Such findings for CSAM IN coatings were also reported by Wu et al. [79]. Nonetheless, this formation was most evident with the He deposit (Figure 8b), of which a nearly densified surface was formed. As such, it was found through EBSD measurements that the He specimen also had the greatest amount of nano-sized grains, which suggests that greater plastic deformation took place. Additionally, due to the extreme plastic deformation, a lesser degree of micro-pores was found measuring at 0.21 ± 0.05%. In contrast, the N_2_ substrate had a porosity of 1.82 ± 0.46%.

Although there is some optimization of processing parameters for pre-processed CS Ni superalloys, many opt for in situ/post-processing techniques to improve their structural/microstructural properties. The advantage of utilizing such techniques will allow for the mitigation of typical defects (porosity, brittleness, etc.) for Ni superalloys and improve their mechanical-, tribological-, and corrosion-based properties. Lou et al. [80] are among the few who demonstrated an alternative way to form fully dense IN718 CS deposits by using in situ (i.e., interlayer deformation) processing through micro-forging (MF). Particularly, this was achieved by applying a mixture of larger-sized 410SS balls during the CS process, which can effectively peen the intersurface of the CS substrate. It was observed that in conjunction with the hammering effect of the IN718 particles, the stainless-steel balls allowed for a severe degree of plastic deformation, which eliminated the possibility of contaminations, oxide scales, and porous defects along the interparticle boundaries. Additionally, it was found that by increasing the amount of MF particle volume (from 25% to 75%), the severe deformation broke the established dendritic framework, thus greatly refining the microstructural features. TEM imaging also supports this observation, depicting nano-sized interparticle gaps, as shown in Figure 9a–d.

With in situ techniques showing desirable outcomes, they are still largely understudied, as the aforementioned work is the only one that investigated in situ techniques. It can be speculated that by utilizing other in situ techniques (e.g., laser melting [81]), a denser and more robust component can be fabricated. However, despite this, many tend to opt for post-treatments to improve the splat’s mechanical/interparticle bonding characteristics. Bagherifard et al. [82] were able to study this by subjecting highly dense freestanding CSAM IN 718 specimens to heat treatment (HT). Specifically, two HTs were conducted on CS specimens, holding at 1050 °C for 3 h and 1200 °C for 1 h. It is typically well-known that the application of HT will promote diffusion and recrystallization, thus allowing for greater interparticle bonding and elasticity [10]. As to be expected, the Inconel samples were effectively densified due to the sintering-like effect of the HT. From a microstructural point of view, the original dendritic structure of the IN 718 sample experienced a sintering-like effect, thus resulting in a more uniform structure (due to recrystallization).

Lastly, other similar post-treatments aside from traditional HT were also utilized for IN coatings. This was shown in the work of Perez-Andrade et al. [83], of which three annealing treatments were applied for IN 718 coatings. These treatments consisted of hot isostatic pressing (HIP), soft annealing, and aging, of which one set of samples was subjected to HIP, whereas another set was subjected to soft annealing and aging. After the HIP annealing treatment, the microstructure consisted of a combination of equiaxed grains, along with δ-phase and MC-carbide precipitates of different sizes and shapes along the grain boundaries. In the case of the annealed and aged specimen, carbides at the particle–particle interface and δ-phase precipitates were found. The presence of recrystallization twins in almost all grains was visualized. However, when their porosities were contrasted, it was found that at a certain point, the solution + aging treatment was able to produce similar porosities and microstructural features to the HIP treatment. As such, it was concluded that HIP treatment is not necessarily mandatory to form defect-free Ni-based superalloy CS components.

The key findings from this section are summarized in Figure 10. In addition works, many other works also detailed similar findings. A summarization of these other works and the list of key observations in relation to their structural quality (i.e., decrease in porosity) before and after post-heat treatments is shown in Table 1. It should be mentioned that to date, there are no existing works that focus on other non-related post-treatments for structural/microstructural augmentation. These treatments, such as laser shock peening (LSP), ultrasonic nanocrystal surface modification (UNSM), and ultrasonic surface rolling process (USRP), can have large implications for augmenting the meso- and micro-scale features of CS Ni-based superalloys [84,85,86,87]. Similarly, from a composite standpoint, there are no existing works that focus on the addition of metal, ceramic, or carbon-based (i.e., graphene) mixtures to the CS fabrication of Ni-based superalloys, except for the work from Sun et al. [88]. In their work, graphene nanoplates (GNPs) were successfully added to IN 718. However, this success was limited due to the poor deposition efficiency (DE) of the GNPs as a function of the GNP content. The reason for this finding was largely due to the particle rebounding effect due to the weakened bond strength of the GNP upon particle impact. That being said, although there is no existing work for harder materials, such as ceramics, it can be assumed that a tampering-like effect would likely occur, which can reduce any type of porous defects. This is, of course, under the assumption that the process is optimized [62]. On the other hand, there can also be a likely effect of surface erosion, which can also result in additional deposition defects [89]. Nonetheless, these works act as a critical underlying segway to elucidate their operational performances in mechanical and tribological applications, which are discussed in the following section.

## 3. Structure–Property Relationships

### Mechanical Properties

When considering the structural augmentations of CS Ni-based superalloys, their formation, whether it be pre- or post-processing will influence their mechanical properties. Under typical CS conditions, the severely deformed particles can present various mechanical advantages and disadvantages. From one perspective, the refined crystalline structure can result in enhanced surface hardness, thus assisting with resistance to plastic deformation. On the other hand, due to the brittle-like nature of the deposit, the intrinsic porous defects can result in pre-mature brittle fracture under tensile/compressive loading conditions. Ma et al. [78] were able to demonstrate this in their work, in which nitrogen and helium as propellant gases were employed for CS IN 718 fabrication. They observed improved micro-hardness with the rise in nitrogen gas pressure in as-sprayed conditions due to increased work hardening and fewer pores in the resultant coatings. When compared with the He-propellant-based sample, their hardness was reduced due to the greater molecular weight. Under tensile conditions, the tensile strength of coatings deposited with He was 400 MPa, which was significantly higher than for N_2_. This was due to the increased particle acceleration, which improved the interfacial bonding. For the N_2_-formed deposit, the fracture morphology indicated that cleavage failure occurred along the non-cohesive splatted particles, as shown in Figure 11a. Such morphologies are quite common with CSAM coatings, as the failure initiates along the interparticle boundaries of the flattened particles [82,97,98]. However, for the He-based specimen, dimples were formed along the fractured surface (Figure 11b). Such a formation can be attributed to the greater particle cohesion from the higher particle impact velocity.

However, as aforementioned, the application of HT can effectively improve the ultimate tensile strength (UTS) and elongation of CSAM Ni-based superalloys (e.g., IN 718) due to the increase in particle cohesion from interparticle diffusion and recrystallization. Mechanistically, this phenomenon is shown below in Figure 12 [99]. That being said, although CSAM parameters can be optimized without post-treatments, the porous/brittle defects of standard CSAM substrates will always exist. As such, by applying such treatments, their mechanical strength can be improved. In fact, relative to other manufacturing techniques, such as traditional casting and fusion-based techniques (i.e., PBF and DED), HT can produce similar-to-superior mechanical properties depending on the processing parameters, as explicated by Bagherifard et al. [96]. This is due to the diffusion-like effect that was previously mentioned, which can effectively improve its ductility. When compared with the aforementioned manufacturing techniques, key mechanical properties, such as ductility and elongation, can suffer due to the precipitation of γ’ (Bi3(Al,Ti)) and γ″(Ni3Nb) alongside δ phases that are a needle-like shape. Due to this, HT CSAM Ni superalloys do fit the requirements of standards such as AMS 5662 and AMS 5383 [96,100].

To further elucidate, Wong et al. [91] exhibited such findings by effectively improving the UTS and elongation properties of CS-+-HT-deposited IN 718 on an aluminum substrate. Compared with its heat-treated counterpart, the as-sprayed coatings exhibited poor ductility. However, when the coating was subjected to HT, the coating became more ductile, which again was attributed to the increase in metallurgical bonding. Further evidence of this can also be seen in Table 2, where a summary of the changes in porosity, hardness, UTS, and elongation among various sources in the literature are listed. Nonetheless, the fractured surface of the CS + HT specimen indicated dimples, which resulted in ductile fracture characteristics. Although not fully explicated, it can be insinuated that these dimples were likely due to the formation of carbide precipitation at higher annealing temperatures (which are often near the material’s melting point) [102,103]. Principally, these precipitates allow for a dislocation-agglomerating-like effect during mechanical action. From an elongation standpoint, these sites act as stress-raisers, which can reduce factors such as elongation, UTS, and impact toughness. Similar findings with other HT technologies (which, in this work, was local induction heat treatment) were also reported by Sun et al. [104], of which the fine presence of δ precipitates acted as prime contributors to flexural failure mechanism, similar to the referenced furnace HT specimen. For reference, flexural failure (i.e., flexural fracturing) occurs after a material yields during a flexural (i.e., bending) test. The fracture morphology of these studied specimens is shown in Figure 13.

One additional point that should also be considered in this discussion is the change in hardness from CS substrates relative to other AM processes, both in their pre-processed and post-processed states. Bagherifard et al. [96] are among the only ones who conducted such an extensive study. In their work, IN 718 was fabricated using CS and SLM techniques. For the CS sample, it was HT at 1050 °C (CS-HTA) and 1200 °C (CS-HTB). In the case of the SLM sample, due to the similar mechanical properties of the aforementioned HT temperatures, only the 1200 °C HT temperature (SLM-HTB) was used. For further comparison, two additional heat treatment strategies for the SLM substrate were used. For the first one (SLM-HTC), a maximum temperature of 980 °C, followed by a heating/holding/cooling cycle over a 24 h period was used (as per AMS5662 [105]). For the second substrate (SLM-HTD), a 980 °C temperature was applied for 1.5 h. As such, their corresponding hardness characteristics are shown in Figure 14. It can be seen that in their as-processed conditions, the CS substrate had the highest hardness relative to the SLM specimen. This explanation can be attributed to the high degree of dislocations of the CS substrate (due to the severe plastic deformation of the particles) relative to the melting and solidification process (which can cause grain growth) for the SLM substrate. However, upon HT, the hardness of the CS substrates significantly decreased due to the softening effect of the heat treatment. However, for the SLM substrates, their hardness significantly increased due to the solution strengthening and precipitations that formed. Although the hardness for the casted IN was not studied, the authors mentioned that typical values lay between 230 HV and 380 HV. It should also be mentioned that there are no existing works that compared the hardness of pre-processed and post-processed CS Ni-superalloy substrates to those fabricated using DED techniques. However, it can be speculated that similar findings would be found.

Collectively speaking, although HT is a viable technique for improving the mechanical properties of CS Ni-superalloy substrates, it often comes at the expense of diminished hardness. To overcome such obstacles, it is suggested that post-deformation treatments, such as LSP and friction stir processing (FSP), be used to improve these properties. To date, there are no existing studies that focus on this topic. As such, the authors suggest that this notion be further studied to advance the field. By studying the effects of alternative post-treatments, the mechanical properties of CSAM Ni-based superalloys can be effectively tailored.

## 4. Tribological Properties

Up to this point, it can be seen that there is a clear influence of the structure on the properties of pre- and post-processed CS Ni-based superalloys. Aside from traditional mechanical tensile and compressive performances, another form of mechanical degradation is tribological interactions. From an industrial perspective, such interactions for Ni-based superalloys largely occur in moving MMAs based on high-temperature aerospace applications [106]. To date, all works focused on this subject tended to study the effects of high-temperature tribological interactions due to the unique formation of protective surface oxides (which can act as a solid lubricant) during abrasion [106,107,108,109]. Although post-treatments, such as HT, can be utilized for the CSAM of IN 625/718 substrates, the diffusion and grain growth that occurs reduce their mechanical hardness in comparison to their as-built counterparts. Such a reduction can often be a detriment in environments that involve tribological abrasion [110]. As aforementioned, it can also be speculated that other deformation-based post-treatments can result in increased wear resistance due to the improvement in surface hardness. Similarly, post-treatments can also assist with the structural integrity of the deposit, as less brittle fracturing can occur during triboloading. However, such works do not exist at this current time. Nonetheless, contemporary works that focused on the tribological mechanisms of CS Ni-based superalloys are covered in order to establish and understand their core tribological mechanisms.

Sun et al. [109] are among the few who studied the tribological response of CS IN718 coatings in high-temperature (up to 600 °C) environments. To purpose of this work was essentially to understand the tribofilm-forming mechanisms of CS IN718 as abrasion at high temperatures is performed. In this work, the methodology was initiated with the deposition of the CS component at a process gas temperature of 1000 °C and a pressure of 4.5 MPa. The coatings were then tested in an air atmosphere at temperatures of 100 °C to 600 °C (in increments of 100 °C). According to their findings, the formation of oxides became increasingly prevalent along the surface as the temperature increased. As shown in Figure 15a,b, Raman spectra were acquired from the coating surfaces to effectively analyze the surface oxides. After thermal exposure to air at 600 °C, the NiFe_2_O_4_ spinel phases were easily detected on the sample surface (Figure 15a). The formation of this phase was attributed to the high concentrations of Ni and iron in the IN718 coatings (Figure 15c). In this research, the authors also speculated that the external energy (at temperatures lower than 400 °C) was below the activation energy needed for starting chemical reactions and could not cause the oxidation process. Under tribological loading, many areas of the formed oxide layer began to delaminate, leaving oxide debris along the wear track. Over time, the debris accumulated and compressed into a new and robust stable oxide layer. This debris could act as a lubricant, thus controlling the friction and wear response of CS IN718 coatings in high-temperature wear environments, as shown in Figure 15b,d,e. Closer inspection along the wear track cross-section also confirmed that the degree of CS splat debonding decreased as a function of temperature, which was attributed to the load-bearing capabilities of the glazed oxide layers. Per Figure 15a, the oxide films began to form on surfaces above 200 °C with sliding. However, the presence of a NiFe_2_O_4_ spinel was clearly identified on the wear track of the sample at 600 °C like the observed phase on an IN718 coating surface after oxidation (at 600 °C without sliding). Rodriguez et al. [111] reported the NiFe_2_O_4_ spinel formation on an IN718 surface when it is exposed to elevated temperatures. The presence of chromium oxide underneath the NiFe_2_O_4_ spinel oxide layer was also postulated by J. Kim et al. [112]. This could prevent the oxygen from further inward penetration into the alloy. Furthermore, the formation of NiFexCr_2_-xO_4_ spinels on the wear tracks of the IN718 coatings was reported to account for the observed low COF values at high temperatures (Figure 15d). Interestingly enough, dynamic recrystallization was also reported beneath the glaze layer due to the combination of plastic strains and stress transfer from repeated load cycling. As a consequence, the surface was hardened, thus providing additional resistance to particle delamination and wear.

Padmini et al. [113] also observed similar findings in their work studying the high-temperature tribological mechanisms of CS IN 615 alloy. Using a lower gas pressure (3 MPa) and temperature (800 °C), the CS coatings were deposited onto SAE 213 TSS and T22 boiler steel substrates. Testing in both room and elevated temperatures (200 °C, 300 °C, and 400 °C), the formation of protective oxide layers along the surface layer helped to reduce the degree of mechanical abrasion along the surface. However, the differentiation in this work lay in the explanation of the formation of the oxide layer, as the authors attributed this formation as also being influenced by the elemental composition of the coating. In this case, the authors attributed the rapid formation of oxide layers due to the concentration of niobium, molybdenum, and chromium along the surface. The wear tracks and their elemental analysis (alongside the CS XRD spectra) are shown in Figure 16a–f. Considering these findings in relation to the work of Sun et al. [109], there are a few takeaways that can be made. First, the primary mechanism for the high-temperature lubrication is due to the formation of the NiO phase at elevated temperatures. Due to the general composition of Ni-based superalloys, the combination of nickel oxides, as well as chromium oxides (i.e., Cr_2_O_3_), allows for a solid-like film that can act as a barrier against abrasion [114]. This would explain the smoothened wear tracks and reduced wear/COF seen in Figure 15d,e and Figure 16a,b, especially considering the porous defects of CS (which can oftentimes result in premature brittle fracturing under abrasion [115]), where these glaze-like characteristics can be of great advantage in such high-temperature environments.

However, for IN 625 deposited in room temperature conditions, Wu et al. [75] reported that the behavior of the oxide-rich tribofilms responded differently with increased material wear at increasing velocities (2 cm/s, 4 cm/s, 6 cm/s, and 8 cm/s) and loads (2 N and 5 N). According to their findings, any increase in load and/or velocity resulted in rapid frictional heating. Although there was the formation of a protective oxide layer because of the heat, it was not as sufficient as the ones typically formed in high-temperature environments. In fact, the oxide wear debris formed from film delamination (due to the presence of Hertzian shear stresses exceeding the shear strength of the surface) fragments into small pieces that had not sufficiently penetrated the surface. These small fragments then acted as third-body wear mechanisms, resulting in greater abrasion along the surface. In combination with the thermal softening that occurred, greater wear rates were observed, despite the formation of the protective oxide film.

Other works from Wu et al. [79] also came to similar conclusions. In this work, the tribological performance of the CS IN 713C deposit was compared with bulk IN 718 in dry room temperature conditions. Having been tested under the same triboloading conditions (ball-on-disk test with a 100Cr6 counter ball), it was found that the CS IN 713C coating had a lessened wear rate compared with the bulk IN 718 materials. The authors elucidated these findings and attributed the decreased wear rate to the greater presence of metal oxides along the surface due to the refined grains of the CS deposit. Similar to the aforementioned work, these tribofilms were consistently delaminated and re-formed throughout the wear track. As such, when compared with the bulk substrate, there were no shear bands along the wear tracks, indicating a lesser degree of abrasive wear. For reference between CS 713C [79] and CS 718 [75] in triboloaded room conditions, Figure 16a–h depicts the difference in wear track morphologies of the CS IN 718 deposit with sliding velocities of 2 cm/s (Figure 17a,b), 4 cm/s (Figure 17c,d), 6 cm/s (Figure 17e,f), and 8 cm/s (Figure 17g,h). Figure 17i,j depicts the wear track morphology of the CS IN 713C deposit as well. Overall, the commonalities between these works lie in the lackluster formation of the oxide film due to the frictional heat of the dry sliding conditions. Although a tribofilm can occur, it tends to be weak and result in a stronger degree of abrasive and adhesive wear. Nonetheless, when compared with traditional as-casted specimens, the CS fabricated samples still had a lower wear rate due to the hardened surface from the CS deposition.

Delving more into the tribological responses of CS Ni-based super alloy deposits, Cavaliere et al. [116] also identified the fretting-based mechanisms of CS IN625 deposit. In contrast to traditional sliding wear, fretting is another area of tribology, which focuses on the effects of low-amplitude tangential cyclic stresses along the surface. Having used normal loads of 50, 100, and 150 N with a 0.09 mm sliding distance, it was interestingly found that the fretting wear decreased as a function of the load (Figure 18a). Although the authors only attributed the decrease in wear loss to the hardness of the underlying material, it can be speculated that the frictional heat generated during fretting enabled the formation of a protective tribo-film along the affected region, as the COF was gradually decreased as well (Figure 18b). It can also be potentially speculated that the wear debris generated from the fretting process also work-hardened the CS coating, which was attributed to the decreased wear volume. For reference, optical images of the fretted regions are shown in Figure 18c–e. However, additional studies are needed to fully explain this phenomenon.

With these mechanisms in mind, it is also important to briefly understand the variations of other AM techniques for wear-resistant Ni-based superalloys and whether CS technology provides any type of advantage. To date, there are no existing works that do such a comprehensive comparison. However, CS is known to have higher hardness compared with other additive/traditional manufacturing methods. This is largely due to the severe plastic deformation induced by the deposition process, as aforementioned [117]. Similar to the discussion in the previous section, Bagherifard et al. [82] were able to demonstrate this in their second work comparing the mechanical hardness of IN 718 fabricated by selective laser melting (SLM) and CS. As to be expected, the CS specimen demonstrated a higher hardness (being at 540 HV compared with SLM at 300 HV) solely due to the immense compressive residual stresses of the deposition process. For reference, traditionally manufactured (i.e., casting) IN 718 is reported to have a hardness of 281 HV [118]. When relating Archard’s theoretical equation of wear [117], it can be suggested that the CS sample would theoretically have greater wear resistance. Although the mechanisms of such processes are much more complex than this simple relation, the fundamental idea is that it gives a sense of how advantageous CS technology is, even in its as-processed state. Altogether, when reflecting on this overall discussion, it can again be suggested that with the use of deformation-based post-treatments, the tribological resistance of these components can be significantly improved, just as it was reported for other AM processes for different materials [119,120]. By doing so, the common outcome of refined microstructural features will effectively improve their tribological performance. Overall, the authors believe that this new sector of research will expand the current understanding of CS Ni-based superalloys.

## 5. Summary and Future Outlook

In recent years, the application of AM Ni-based superalloys has become increasingly popularized due to their significance in high-temperature applications, such as gas turbines and nuclear reactors. As such, there is a large reliance on fusion-based AM techniques for Ni-superalloy fabrication. Although these techniques have their respective advantages, all of them tend to be reliant on thermal-based processes, which can lead to oxidation, undesirable phase transformations, hot tearing, cracking, thermal residual stresses, and alloying elemental segregation. One alternative AM technique that counteracts all of these disadvantages is CSAM technology. By utilizing the kinetic energies of rapidly accelerated particles through a portable nozzle, CSAM acts as a rapid and reliable solid-state technique for Ni-based superalloy fabrication. Due to its solid-state-like advantages, CSAM Ni superalloys have been applied in various fields, such as aerospace, automotive, and nuclear fields.

To understand the viability of CSAM for Ni superalloys, the bonding mechanisms were first investigated. It was found that CSAM is indeed an effective deposition technique for Ni-based superalloy powder particles due to the mechanical interlocking of the particles upon impact. With the recrystallization mechanism being controlled by CDRX, there is great feasibility for the CSAM process. However, the presence of brittleness and porous defects can largely hamper the mechanical characteristics of CS-formed Ni-based superalloys due to their tendency for brittle fracturing; in particular, when compared with fusion-based techniques, due to the severe work hardening of the CSAM process, key factors such as UTS and ductility are inferior compared with other AM processes. To combat such defects, post-cold spray HT is among the most commonly used methods for improving the mechanical properties of Ni superalloy CS deposits. The concluding mechanism largely relates to the diffusion that occurs throughout the interparticle boundaries during the HT process. However, despite this advantage, undesirable precipitates can occur, which can act as stress raisers during mechanical loading. From a tribological perspective, the formation of robust oxide layers along the surface was reported for most CS Ni-based superalloy deposits at elevated temperatures. This could act as an effective lubricant against mechanical abrasion. However, despite this knowledge, there still exist many gaps in the knowledge pertaining to the application of other post-treatments and how they would affect their structural, mechanical, and tribological performances. Specifically, it is highly suggested that deformation-based techniques be investigated as a method for improving the surface properties of CSAM Ni superalloys. By doing so, it can be suggested that the strengthened surfaces will allow for greater degradation resistance in a multitude of environments. Similarly, there are other properties outside of mechanical/tribological analysis that should be further investigated to progress in the field, such as corrosion and high-temperature oxidation mechanisms. However, despite these gaps in the field, the primary takeaway is that the CSAM process is indeed a suitable process for Ni-based superalloys. Stemming from the ease of deposition, impressive tribological performance, and suitable mechanical characteristics (via techniques such as HT), there is great potential for this process for various high-temperature applications. Collectively, we believe that by realizing this knowledge, the preservation of various mechanical-based types of machinery can be preserved, thus greatly extending their lifespans for continuous application.

## Figures and Tables

**Figure 1 materials-16-02765-f001:**
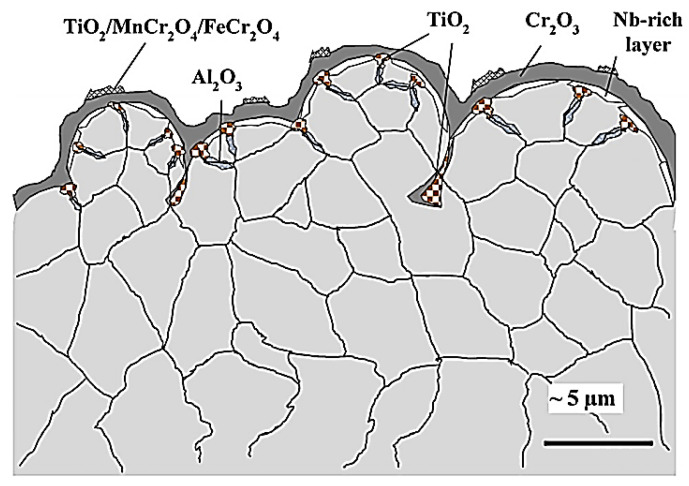
Formation of surface oxides on an additively manufactured IN 718 surface after high-temperature oxidation. Reprinted from Sanviemvongsak et al. [31], Copyright 2018, with permission from Elsevier.

**Figure 2 materials-16-02765-f002:**
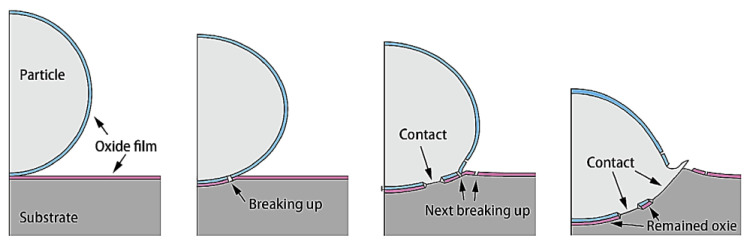
The breakup of the oxide surface shell upon the impact of powder particles in the CS process. Reprinted from Ichikawa et al. [51], Copyright 2018, with permission from Elsevier.

**Figure 3 materials-16-02765-f003:**
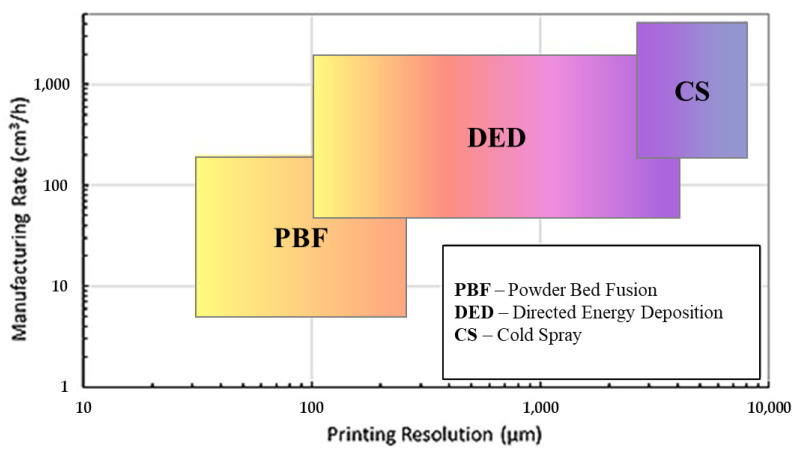
The relationship between the manufacturing rate and printing resolution for cold-spray- and laser-based AM processes. Adapted from Zou [54], Copyright 2021, with permission from the American Chemical Society.

**Figure 4 materials-16-02765-f004:**
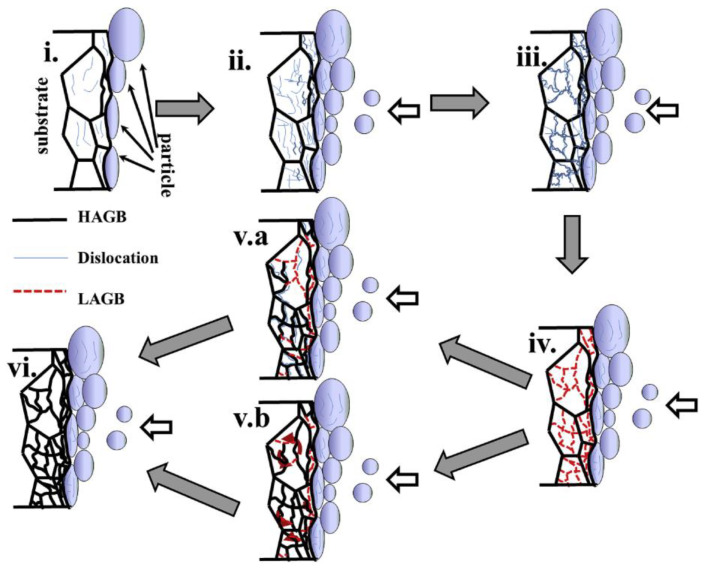
Schematic illustration of the microstructural evolution at the steel substrate near the interface between a CS coating and substrate: (**i**) deformation near the interface, (**ii**) dislocation accumulation at the interface, (**iii**) dislocation tangling and formation of cells, (**iv**) rearrangement and subgrain development, (**v.a**) HAGB formation due to dislocation accumulation from an LAGB, (**v.b**) an HAGB formed due to CDRX, and (**vi**) final microstructure of substrate are represented. Reprinted from Chaudhuri et al. [65], Copyright 2017, with permission from Elsevier.

**Figure 6 materials-16-02765-f006:**
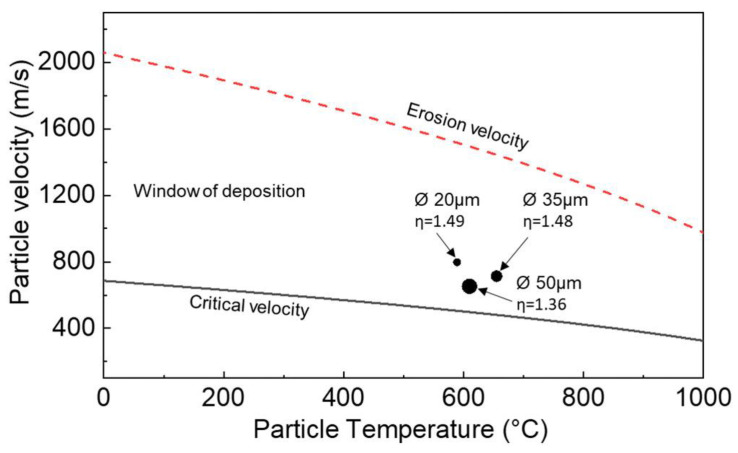
A visualization of the window of deposition of IN 625 alongside the particle velocity of various particle sizes. Reprinted from Wu et al. [75], Copyright 2021, with permission from Elsevier.

**Figure 7 materials-16-02765-f007:**
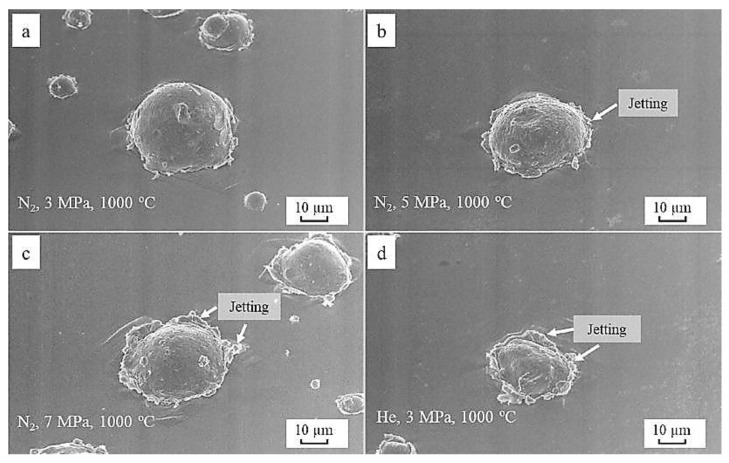
The morphology of a single splat of IN 718 on the substrate deposited using a constant temperature at 1000 °C and nitrogen at (**a**) 3 MPa, (**b**) 5 MPa, and (**c**) 7 MPa, as well as (**d**) He at 3 MPa. Reprinted from Ma et al. [78], Copyright 2019, with permission from Elsevier.

**Figure 8 materials-16-02765-f008:**
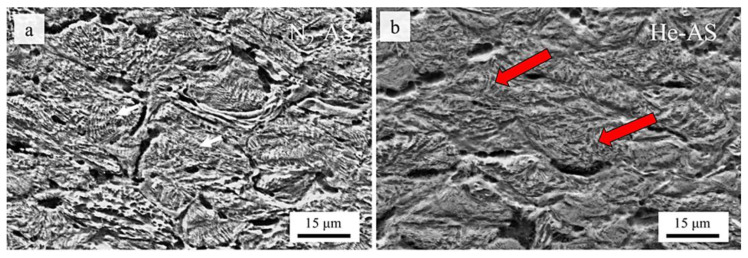
Cross-section morphologies of CS-deposited IN 718 (**a**) sprayed with nitrogen propellant at 7 MPa and 1000 °C and (**b**) sprayed with helium propellant at 3 MPa and 1000 °C with the red arrows indicating the presence of a dendritic-based structure. Adapted from Ma et al. [78], Copyright 2019, with permission from Elsevier.

**Figure 9 materials-16-02765-f009:**
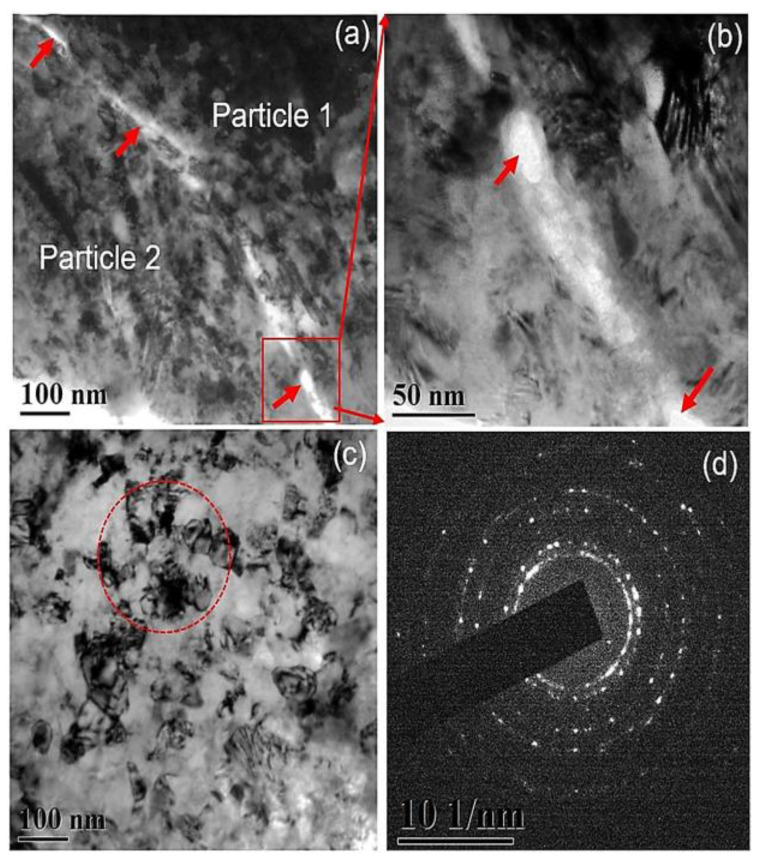
TEM images of the cross-section of a CS + MF IN 718 deposit: (**a**–**c**) nano-sized interparticle gaps and (**d**) supporting selected area electron diffraction (SAED) of the observed region. The red arrows indicate the presence of nano-sized inter-particle gaps, whereas the red circle indicates the refined crystals adjacent to the inter-particle boundaries. Reprinted from Lou et al. [80], Copyright 2018, with permission from Elsevier.

**Figure 10 materials-16-02765-f010:**
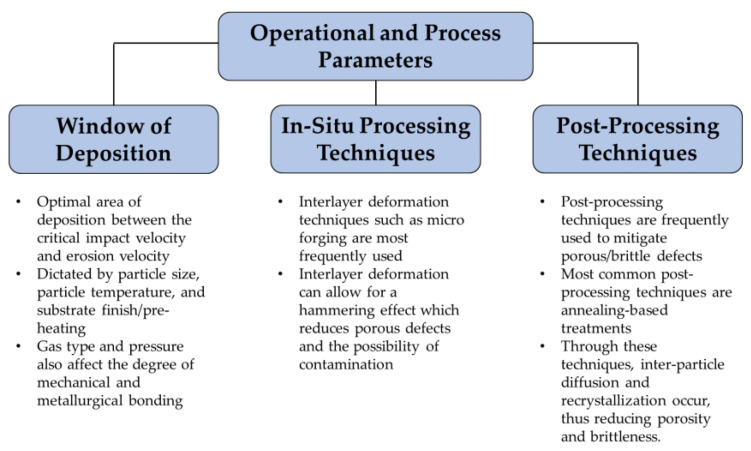
A schematic representation of the key operational and process parameters for CSAM Ni-based superalloys.

**Figure 11 materials-16-02765-f011:**
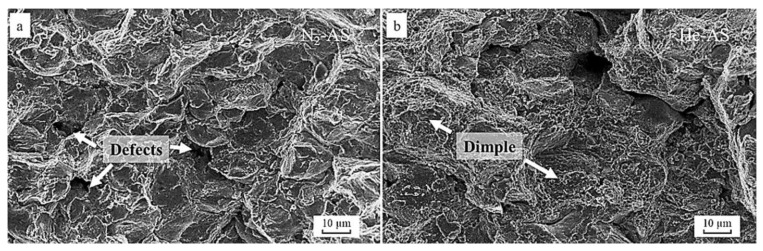
The fracture morphology of (**a**) nitrogen- and (**b**) helium-fabricated CS IN 718 deposits. Reprinted from Ma et al. [78], Copyright 2019, with permission from Elsevier.

**Figure 12 materials-16-02765-f012:**
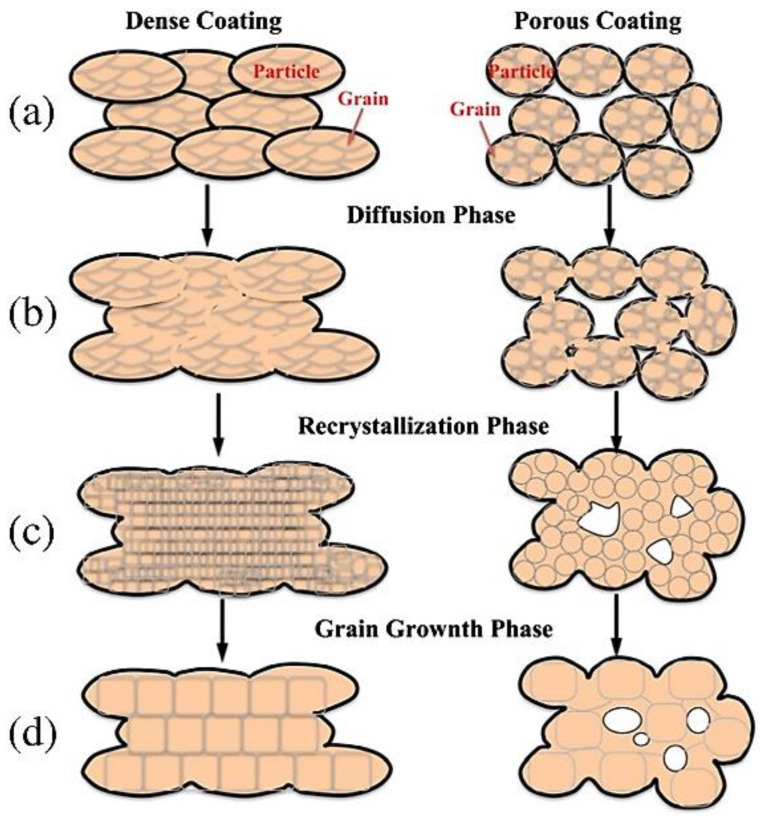
Schematic illustration of the CS deposit structure evolution during post-cold spray heat treatment. The evolution of CS densification is depicted by the (**a**) initial CS deposit, (**b**) the diffusion phase, (**c**) the recrystallization phase, and (**d**) the grain growth phase. Reprinted from Sun et al., Copyright 2020 [101], under CC BY 4.0.

**Figure 13 materials-16-02765-f013:**
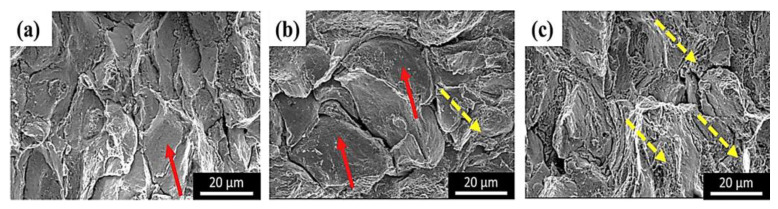
The flexural fracture morphology of (**a**) as-sprayed, (**b**) furnace-heat-treated, and (**c**) local-induction-heat-treated CS IN 718 substrates. The red arrows indicate brittle facets whereas the yellow dash arrows indicate intergranular fractures. Reprinted from Sun et al. [104], Copyright 2019, with permission from Elsevier.

**Figure 14 materials-16-02765-f014:**
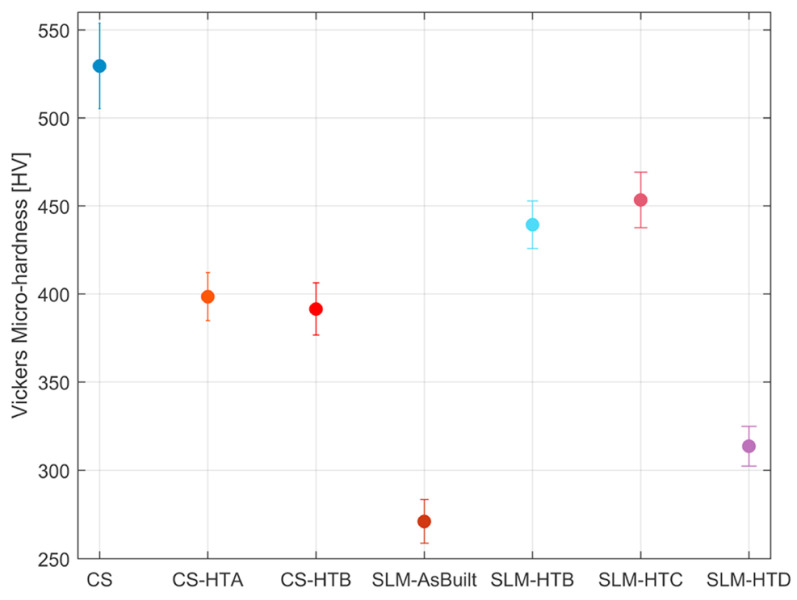
A comparative viewing of the micro-hardness measurements from pre- and post-heat-treated CS and SLM IN 718 substrates. Reprinted from Bagherifard et al. [96], Copyright 2018, with permission from Elsevier.

**Figure 15 materials-16-02765-f015:**
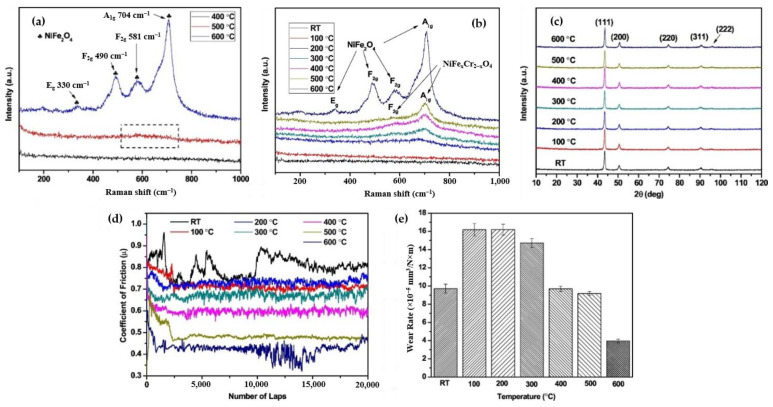
(**a**) Raman spectra of IN718 coatings exposed at different temperatures up to 600 °C, (**b**) Raman spectra of IN718 coatings (after wear tests from low to high temperatures), (**c**) XRD scans of IN718 coatings (in as-sprayed condition and oxidized at different temperatures), and the (**d**) frictional and (**e**) wear responses of CS IN718 in various temperature environments. Reprinted from Sun et al. [109], Copyright 2020, with permission from Elsevier.

**Figure 16 materials-16-02765-f016:**
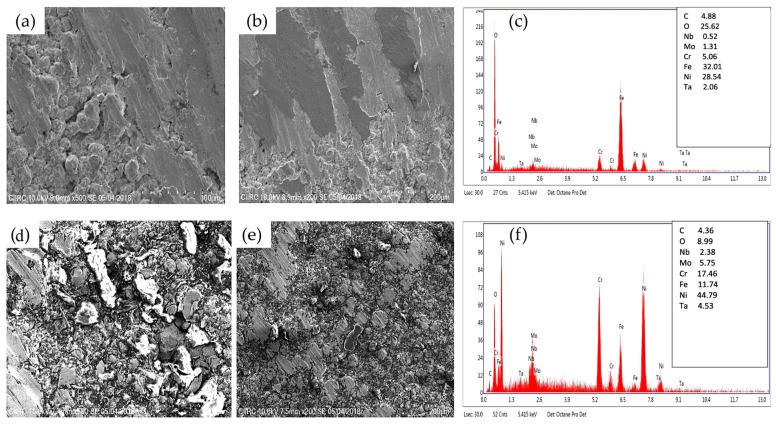
The wear track morphologies of CS IN 625 on SAE213 T11 at (**a**,**b**) 400 °C and (**d**,**e**) 200 °C, along with their elemental composition (**c**,**f**). Reprinted from Padmini et al. [113], Copyright 2020, with permission from Elsevier.

**Figure 17 materials-16-02765-f017:**
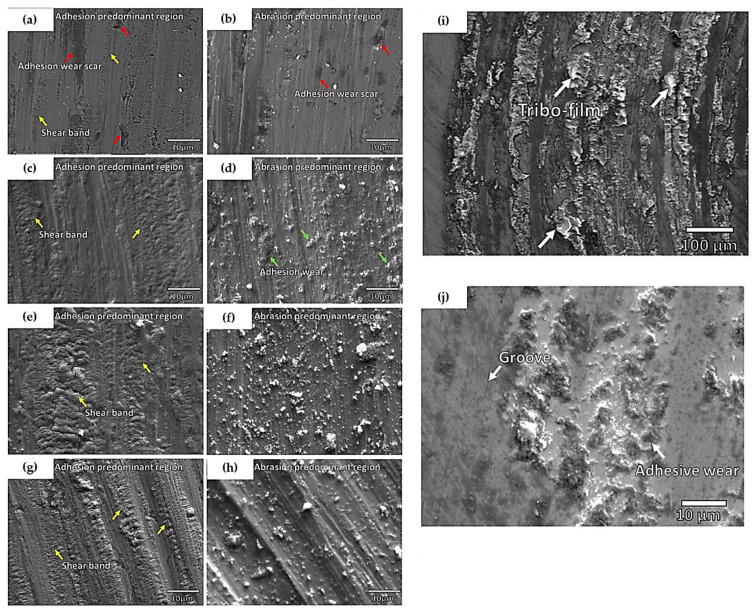
The wear track morphologies of CS IN 718 deposit subjected to dry room temperature conditions at a load of 5 N and sliding velocities of (**a**,**b**) 2 cm/s, (**c**,**d**) 4 cm/s, (**e**,**f**) 6 cm/s, and (**g**,**h**) 8 cm/s. Additionally, the wear track morphology of CS IN 713C deposit (**i**,**j**) is shown. In (**a**–**h**), the yellow arrows indicate shear bands, whereas the red and blue arrows indicate abrasive wear and adhesive wear. In (**i**,**j**), the white arrows indicate grooves, adhesive wear, and the formation of tribo-films. All images are reprinted from Wu et al., Copyright 2021 [79], under CC BY 4.0.

**Figure 18 materials-16-02765-f018:**
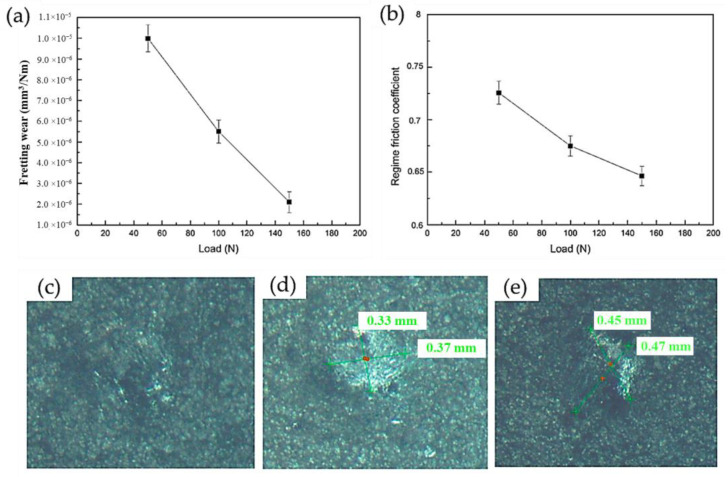
The (**a**) fretting wear; (**b**) COF; and OM of the wear tracks of CS IN625 deposit under (**c**) 50 N, (**d**) 100 N, and (**e**) 150 N loadings. Reprinted from Cavaliere et al., Copyright 2021 [116], under CC BY 4.0.

**Table 1 materials-16-02765-t001:** A summarization of the current literature on the structural and microstructural formation of CS-based Ni superalloys.

Feedstock	Substrate	Spray Parameters	Observations	Porosity % before Post-Treatment	Porosity % after Post-Treatment	Ref.
IN 718	Low-carbon steel	Nitrogen was employed as gas 3.5 MPa gas pressure was used 800 °C gas temperature	Cold spraying showed a lower porosity level but lacked strength Pressureless sintering of the deposits increased strength	2.5% in pre-processed state	Sintering at 1200 °C—1.4% Sintering at 1250 °C—0.2%	[90]
IN 718	Al	Nitrogen and helium were employed as gases 2 and 5 MPa gas pressure was used 1000 °C gas temperature	Higher particle velocity resulted in a denser coating Post-heat treatment of the cold-sprayed sample at a higher velocity resulted in bond strengthening	2.7% at a particle velocity of 787 m/s 3.4% at a particle velocity of 741 m/s	2.0% when heat-treated (1250 °C) and particle velocity of 787 m/s 3.8% when heat-treated (1250 °C) and particle velocity of 741 m/s	[91]
IN 718	Al	Particle size—10 µm to 32 µm Propellant gas—nitrogen Temperature—1000 °C Pressure—55 bar Traverse velocity—500 m/s Standoff distance—25 mm	Dense IN buildup was obtained Spray direction did not affect the mechanical properties and microstructural features Limited ductility for CS specimens Heat treatment showed a reduction in porosity and higher interparticle bonding High strength and ductility were obtained after heat treatment High annealing temperature enhanced the ductility, which was confirmed by a high fraction of dimples in fractured specimens	1.3% for longitudinal 1.2% for traverse	0.9% for longitudinal with heat treatment at 1050 °C 1.1% for traverse with heat treatment at 1050 °C 0.9% for longitudinal with heat treatment at 1200 °C 1.0% for traverse with heat treatment at 1200 °C	[82]
IN 718	IN 718	Nitrogen was employed as gas 4 MPa gas pressure was used 800 °C gas temperature	The coating formed was dense but brittle The coating formed had enhanced tensile strength and ductility	0.2–0.5% in as-processed state	N/A	[92]
IN 625	Carbon steel	Nitrogen was employed as gas 5 MPa gas pressure was used 1000 °C gas temperature	High temperature and pressure resulted in lower porosity and enhanced hardness The corrosion properties of the coating improved	>1.0%	N/A	[93]
IN 625	4130 chrome alloy steel	Particle size—5 µm to 50 µm Propellant gas—helium Temperature—500 °C Pressure—30 bar Mean particle velocity—800 m/s to 1000 m/s	The microstructure near the interface contained small grains Splat and splat boundaries contained dislocations due to the deformation from CS Sub-grains were observed inside the splat Inter-splat boundaries consisted of the shear band and high-density dislocations The average crystallite size of the particle dropped to 160 nm	N/A	N/A	[65]
IN 718	IN 718	Powder feed rate—48 g/min Propellant gas—nitrogen Temperatures—800 °C, 900 °C, and 1000 °C Pressure—50 bar Traverse velocity—500 mm/s Standoff distance—20 mm	Sprayed deposit was free from oxidation and phase transformation High compressive stress and hardness were observed due to the peening effect during deposition Solution and aging treatment could reduce the porosity of the deposits Higher gas processing temperature showed lower porosity in the coating The coating quality parameter increased with an increase in gas process temperature Porosity, deposition efficiency, hardness, and residual stress showed good correlations with coating quality parameter	CS at 800 °C, 900 °C, and 1000 °C: 1.8%, 1.5%, and 1.3%, respectively	CS at hot isostatic pressing + solution treatment + aging treatment at gas temperatures of 800 °C, 900 °C, and 1000 °C: 1.7%, 1.4%, and 0.3%, respectively	[83]
IN 625	Aluminum 6061	Particle size—10 µm to 70 µm Propellant gas—nitrogen Temperature—1000 °C Pressure—4.7 MPa Traverse velocity—500 mm/s Standoff distance—30 mm Powder feed rate—48 g/min	High-quality coatings with low porosity levels Fine grain structure at the coating/substrate interface Adhesion strength of 57 MPa reported between coating and substrate Micro-hardness of coatings was higher than bulk IN 625 Wear rates were in the range of mild wear regimes High wear rate and COF were reported at a higher sliding velocity, which was attributed to the weak protection from tribofilm and higher adhesive and abrasive wear Lower COF was reported for tribologically tested specimens with a load below 5 N	~0.3%	N/A	[75]
IN 718	IN 718	Particle size—15 µm to 45 µm Propellant gas—nitrogen Temperature—1000 °C Pressure—4.5 MPa Solution treatment temperature—900 °C, 950 °C, 1000 °C, and 1050 °C	The solution and aging process helped to improve the diffusion of the CS deposits and reduced the porosity level Segregated heavy elements in the inter-dendritic region dissolved in the matrix Carbides and δ-phase precipitates were observed after solution and aging treatment at lower treatment temperatures and whose fraction decreased with an increase in solution and aging treatment Proper solution and aging treatment could provide CS deposits with comparable bulk properties	1.7%	CS after solution and double-aging treatments at 900 °C, 950 °C, 1000 °C, and 1050 °C: 1.32%, 1.29%, 1.13%, and 1.02%, respectively	[94]
IN 625	IN 625	Particle size—100 µm to 300 µm Propellant gas—nitrogen Temperature—1050 °C Pressure—4.5 MPa Standoff distance—3 mm to 40 mm Powder feed rate—5.45 g/min to 92.65 g/min	Average particle velocity decreased linearly Large standoff distance could produce superior coating Low coating porosity and high hardness were observed for a standoff distance of 3 cm Highest average velocity was reported for standoff distance of 8 cm	0.97% and 1.18% at 3 cm and 8 cm standoff distances, respectively, 1 mm away from the coating-substrate interface	N/A	[95]
IN 718	Aluminum	Nitrogen gas was employed at 1000 °C Propellant gas at 55 bar Transverse velocity at 500 mm/s Track spacing at 1 mm Standoff distance at 25 mm Heat treatment at 1050 °C for 3 h and 1200 °C for 1 h in argon atmosphere	Phase was retained during the CS process and after heat treatment CS microstructure exhibited dendritic structure After heat treatment, highly deformed splats were diminished CS substrates exhibited improvement in tensile strength after heat treatment Heat treatment at 1200 °C exhibited superior UTS performance	~1.04%	0.75% and 0.83% at 1050 °C and 1200 °C heat treatments, respectively	[96]

**Table 2 materials-16-02765-t002:** A summary of changes in porosity, hardness, ultimate tensile strength (UTS), and elongation between different cold-sprayed samples of IN 718 produced with N_2_ and He (as the process gases), micro-forging (MF), and various heat treatments [78,80,91,96].

Samples Sprayed	Parameters	Porosity (%)	Hardness (HV)	UTS (MPa)	Elongation (%)	Reference
As-sprayed, 0% MF	700 °C, 25 bar	5.7	400	96.4	0.12	[80]
As-sprayed, 50% MF	700 °C, 25 bar	0.23	500	463	0.48	[80]
Heat-treated, 1200 °C, 6 h, 50% MF	700 °C, 25 bar	0.5	400	1089	6.17	[80]
As-sprayed, He	1000 °C, 30 bar	0.21	600	1168	0.58	[78]
Heat-treated, He, 990 °C, 4 h	1000 °C, 30 bar	0.18	400	1272	9.64	[78]
As-sprayed, N_2_	1000 °C, 50 bar	2.7	NA	277	0.23	[91]
As-sprayed, He	1000 °C, 20 bar	3.4	NA	204	0.18	[91]
Heat-treated, N_2_, 1250 °C, 1 h	1000 °C, 50 bar	2.0	NA	764	24.7	[91]
As-sprayed	1000 °C, 55 bar	1.2	530	713	0.45	[96]
Heat-treated, 1050 °C, 3 h	1000 °C, 55 bar	1.0	400	1260	8.85	[96]
Heat-treated, 1200 °C, 1 h	1000 °C, 55 bar	1.0	400	1289	15	[96]

## Data Availability

Data sharing is not applicable to this article.

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
