# Peer review of "Solid-State Cold Spray Additive Manufacturing of Ni-Based Superalloys: Processing–Microstructure–Property Relationships"

_materials, 2023, doi:10.3390/ma16072765_

Round 1

Reviewer 1 Report

General: The manuscript addressed a meaningful and relevant topic in the AM processes field, reposing the process characteristics and material performance. However, the absence of the authors' critical view is the main weakness of the manuscript. Increase the figure resolution (this link can help: https://www.youtube.com/watch?v=TEFJBeZmsCI&t=253s).

The authors extensively discuss PBF in the introduction, which mischaracterizes the manuscript proposal. Rewrite the introduction addressing why the cold spray AM must be studied, its advantages and limitations in relation to others AM processes, and the main technical problems observed in the literature. What standard classified the cold spray AM ? Why are Ni-alloys suitable to be deposited via cold spray AM?

What is the main advantage of a solid-state process in relation to a fused-based one (e.g., DED and PBF)?

The authors also extensively discuss IN 718 and IN 625 alloys fabricated via L-PBF and their properties (room and high temperature), which defocused the introduction.

The manuscript objective is not clear.

The introduction was chaotic and extensive. Make a focused introduction, addressing the main points of the technology and why it has industrial scalability.

Add a paragraph explaining the review sequence and topic distribution. This increases the manuscript's readability.

The specific discussion about the PBF, Ni-alloys, and hot and cold spray process presented in the 1 Introduction can stay in subtopics or be used to increase the discussion in topic 2.

"These benefits consist of having higher flexibility, no specific limit on the product size, lesser production times, and suitability for the repair of damaged components [62]". DED processes also have similar advantages in relation to PBF.

"Considering that porosity is one concern for CS Ni-based alloys, many have sought to improve their inter-particle boundaries (their quality) through processes such as fine-tuning the highpressure cold spray parameters and post-cold spay heat treatments". Discuss about the use of interlayer deformation processes to reduce the pore size (e.g., https://doi.org/10.3390/app6110304).

The manuscript did not address the results of the Ni-alloys fabricated via DED processes. These are an import class of AM process. Most cited papers talk about the PBF processes, which reduced the manuscript coverage.

Make some industrial examples of the Cold spray. What is the maximum part already built via cold spray AM? Can cold spray concur with DED (both processes had a high deposition rate)?

Considering that the In625 did not have mandatory post-deposition heat treatment, is the cold spray AM suitable for this alloy? The built material will have an intense deformation, reducing corrosion resistance. Also, considering the deposition via cold spray in a steel substrate (e.g., HSLA), the heat treatment temperature (e.g., 980 and 1100 °C, ASM 5662) of the Ni alloys is upper the austenitization temperature, which can jeopardize the toughness of the substrate. Similarity, for Al-alloys. The heat treatment is higher than the melting temperature. Please discuss about these topics.

The authors only described the papers but did not perform a holistic discussion, which critically reduced the novelty of the review. What is the author's point of view about the cited manuscript? Which limits the industrial scalability of the cold spray AM?

Discuss about the recrystallization during the post-deposition heat treatment of the In718 fabricated via cold spray AM. This is a critical point for the In718.

"The tensile properties were poor during as-sprayed conditions and the as-sprayed deposit also showed brittle behavior. However, after heat treatment, a superior increase in ultimate tensile strength and elongation were reported when He was used as propellent gas." This is obvious since the Ma et al. work studied the In718.

Considering the tribological aspects of the material fabricated by cold spray, most of them were used to clad a previous substrate. So, is the cold spray used to clad, additive manufacturing, or both? What are the advantages to used cold spray to build a part? Please clarify these topics.

The wear resistance of the Ni alloys fabricated via Cold spray AM is better, bad, or similar to Ni alloys fabricated via cast, welding, other AM processes, and forged? Please compare them. Also, explain the primary wear mechanism observed for Ni alloys cold strappy AMed.

Please provide the main manuscript conclusion and outlooks on different topics. In addition, given that the manuscript's objective is unclear, it is hard to really understand the conclusion. So, first defines the manuscript's objectives and then make a clear conclusion.

Reviewer 2 Report

The content of this paper is perfect, and the literature analysis and prospect are reasonable.

My specific comments are as following:

1、Some molecular formulas in this manuscript have the problem of inconsistent upper and lower scripts.

2、It should be added that the current application status and deficiency of nickel coating compared with the other AM techniques.

3、The effect of other metal or ceramic powder mixtures on the properties of nickel-based coatings should be briefly described.

4、Some of the images are not clear enough, as shown in figures 16 and 21.

5、It is best to summarize the schematic of the wear mechanism yourself.

Reviewer 3 Report

The idea of the paper is interesting. The paper is too long but good review paper. There are only a few minor suggestions.

1. Equation 1 and its description should be in a separate chapter, not in the Introduction.

2. It will be nice if you can mention other additive manufacturing technologies such as fused deposition modelling (https://doi.org/10.46793/tribomat.2022.009).

3. References 17 and 18 are about aluminium-base MMC, not Ni-based composites. Please use some other reference.

4. The part regarding the Ni-based superalloys description is not necessary and should be deleted from the Introduction.

5. On the other hand, the application of superalloys on a Ni-base in thermal barrier coatings should be mentioned (https://doi.org/10.1179/1743284712Y.0000000193).

6. Figures 1-3 (other researchers' results) could be deleted and only the findings referenced and presented in the Introduction part.

7. Figures  4-6 are not necessary and should be deleted.

8. The text „Although thermal spray coatings may have their own set of defects“ needs a reference, e.g. Vencl, Optimisation of the deposition parameters of thick atmospheric plasma spray coatings, Journal of the Balkan Tribological Association, 18 (2012) 405-414 or Lugscheider et al. Modeling of the APS plasma spray process, Computational Materials Science, 7 (1996) 109-114.

9. Figure 9 could also be avoided.

Reviewer 4 Report

The manuscript shows a very long and extensive effort by the authors on understanding the criticalities and advantages of cold-spray AM on Ni-based coatings.

Although the overall work is scientifically sound, the reviewer feels that the manuscript is more an review than an experimental investigation.

While the manuscript is a review, in my opinion the abstract is misleading, as I think that it seems to focus on an experimental investigation rather than being a review.

The organization of the manuscript is confusing, the introduction is excessively long, with a degree of detail that can be expected from a thesis or a review.

The reviewer suggests clarifying the intentions within the article, that is, choosing between a review and the investigation.

The experimental part, while is pretty descriptive, on the other hand is confusing as the manuscript lacks of a proper “Materials and Methods” part, as well as of a proper discussion.

In general, while the great effort of the authors is clear, the reviewer thinks that the article does not keep what has been promised in the abstract, just because of its chaotic structure.

For this reason, it is suggested that the authors give a clearer structure to the whole manuscript.

Reviewer 5 Report

Dear Authors,

if possible, can you add some examples of real industrial applications of IN625/IN718 cold sprayed coatings? I mean, not only "aeronautical sector", but an example of a specific part. 

Round 2

Reviewer 1 Report

All comments are in the attached file.

Reviewer 4 Report

Clarifications, additions, and modifications done by the authors, together with their accurate responses to the reviewers, are satisfactory.

The manuscript can be accepted in this form.
